# INTERVENTION ADVERSARIAL AUTO-ENCODER

## ABSTRACT

In this paper we propose a new method to stabilize the training process of the latent variables of adversarial auto-encoders, which we name Intervention Adversarial auto-encoder (IVAAE). The main idea is to introduce a sequence of distributions that bridge the distribution of the learned latent variable and its prior distribution. We theoretically and heuristically demonstrate that such bridge-like distributions, realized by a multi-output discriminator, have an effect on guiding the initial latent distribution towards the target one and hence stabilizing the training process. Several different types of the bridge distributions are proposed. We also apply a novel use of Stein variational gradient descent (SVGD) (Liu & Wang, 2016), by which point assemble develops in a smooth and gradual fashion. We conduct experiments on multiple real-world datasets. It shows that IVAAE enjoys a more stable training process and achieves a better generating performance compared to the vanilla Adversarial auto-encoder (AAE) (Makhzani & Shlens, 2015).

## 1 INTRODUCTION

Generative models are widely used for image and texture production. Among them there are two base models which are most appealing to scholars for their elegant theoretical foundation and close combination with neural network, namely Variational Auto-encoders (VAE)(Kingma & Welling, 2013) and Generative adversarial network (GAN)(Goodfellow et al., 2014). VAE maximizes a lower bound of the log-likelihood called ELBO which decomposes to a reconstruction loss term and a regularization term within an auto-encoder structure. GAN, however, goes beyond the likelihood concern, trying to generate data directly using a learned map from the latent space that is trained in an adversarial manner. Both models have some weakness. For example, GAN's training is quite unstable and sometimes may have mode collapse problem. VAE usually produces less sharp pictures. The regularity of the latent space is also an important concern for VAE that affects the quality of the generated images.

There're many attempts to alleviate the above drawbacks. Adversarial Auto-encoders (AAE) combines the techniques of both VAE and GAN network. It imposes a discriminator on the latent space of an auto-encoder to classify the latent distribution $p(z)$ and $q(z)$, where the adversarial term serves as the regularizer. Such choice improves the regularization effect of the latent variable and alleviates the mode collapse problem due to an auto-encoder structure. Wasserstein Auto-Encoders(WAE)(Tolstikhin et al., 2017) generalize different regularization approaches on latent space, proposing adversarial method (WAE-GAN) and kernel based method (WAE-MMD). It provides AAE with theoretical support as well. Intervention Generative Adversarial Network (IV-GAN)(Liang et al., 2020), another attempt to combine GAN with the encoder structure, stabilizes GAN's training process by intervening on the latent distribution so that the reconstructed data distribution could approach the target more robustly.

In this work, we will propose this novel regularization method on AAE models. As basically an auto-encoder structure, the key question is how to let the original distribution in latent space approach the prior distribution. We intervened on the encoded data, which inherently brings intervened distribution. We will show that a loss term in our model can be transformed to a certain measure of the overlap among multiple distributions. When minimizing the loss in terms of these constructed variables, we expect not only the original distribution to approach the target one, but the distribution of the intervened data (we call bridge distribution) to serve as a guidance to have such approaching process more fast and robust as well.

## 2 METHODOLOGY

Training a GAN model has always been a challenge (e.g., the process is known to be unstable and prone to mode collapse). AAE's auto-encoder structure alleviates the problem with a reconstruction term. It first updates the auto-encoder upon the reconstruction loss, then use the discriminator to distinguish the encoded latent variables and generated ones following Gaussian distribution, updating the discriminator and encoder in an adversarial manner. However, such architecture doesn't completely solve GAN's problem. Decreasing the JS divergence has a relatively low approaching efficiency between two distributions. Adversarial training on latent space doesn't avoid this problem at all. Therefore there's a desire to apply a more powerful regularization term to have the approaching process more efficiently and stable. We consider to construct a series of "bridge distributions" by certain transformations so that they in a sense lie between the encoded distribution and target one. We let the discriminator to classify these multiple distributions, and update the encoder to let them approach each other. Intuitively such bridge distributions would serve as a guidance to help the latent variables follow the target distribution more quickly and robustly, but we need more theoretical arguments first.

Before we expound on further details, we want to turn our attention to how to construct the transformations mentioned above. We first explain the meaning of intervention. It is similar to what is exhibited in IVGAN(Liang et al., 2020). For completeness we show it here in our context.

**Definition 1** *Let $\mathscr{X}$ be the set of all random variables in $R^d$ on the probability space $(\Omega, \mathscr{X}, P)$. $\mathscr{T}$ is the set of all mappings from $R^d$ to $R^d$. For a distribution function with support in $R^d$, we call $T \in \mathscr{T}$ "P-intervention" if $X \sim \mathbb{P} \Rightarrow T(X) \sim \mathbb{P}$, for all $X \in \mathscr{X}$. If $S \in \mathscr{T}$ satisfies that $T(X) \sim \mathbb{P}, \forall T \in S \Rightarrow X \sim \mathbb{P}$, we denote $S$ "Complete Intervention Group".*

A rough understanding of the above definition is that complete intervention group let the meet of subsequent multiple distributions embed their final distributing pattern, i.e. $\mathbb{P}(x)$. Combining the idea in our model, the intervention can be performed on our encoded data. We hope when these originally different distributions approach each other, they all approach the prior distribution, which is exactly the feature of the transformation proposed above. Therefore the complete intervention group can serve to ensure the identifiability of our method.

To be more concrete, we set the $\mathbb{P}$ to be standard Gaussian and propose several intervention patterns that possess the above feature.

A simplest example of $\mathbb{P}$-intervention is to replace some of the dimension with normal distributed one. Therefore we have our **Blockwise substitution**: $R^d$ space is divided equally by $t$ parts, where $t|d$. , For $0 \leq k \leq t$, let $T_k$ be the transformation that substitute the first k blocks with random variables following standard normal distribution independent with the original variable. Then we get a complete intervention group $\{T_k\}$.

There's other ways to create intervention groups. For example, we let $x$ be replaced by $xcos(\frac{k\pi}{2t}) + zsin(\frac{k\pi}{2t})$, for $k = 0, 1, ..., t$, where $z$ follows standard normal distribution independent with $x$. We call such construction pattern **Radial substitution**.

Except for the two substitution method, we also apply Stein variational gradient descent (SVGD) (Liu & Wang, 2016) to construct the bridge distribution by iteratively updating points. By setting the target distribution P the updated particles will approximately approach P. More details will be specified in the next section.

Now we propose our model called *Intervention Adversarial Auto-Encoder* (IVAAE). [Figure 1] shows its specific structure, which contains an encoder $E$, decoder $G$ and a discriminator $D$. Images $x$ is fed into the $E$ to get the encoded latent code $z$, which is decoded by $G$ to obtain the reconstructed images $x$. The encoded $z$ is transformed by a complete intervention group $T_k$ ($T_0$ represent the identical mapping) to get $z_k$ which is discriminated by $D$ with output of size $t + 1$. Through maximizing and minimizing alternatively a cross entropy loss, $D$ and $E$ is both updated.

**Intervention Loss**

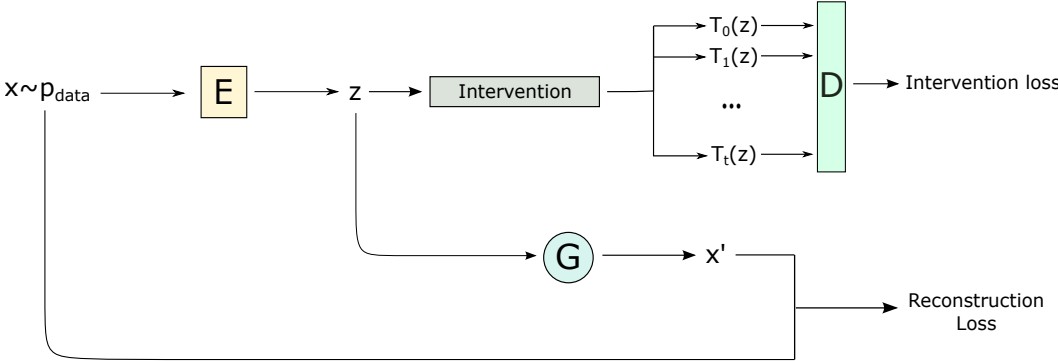

Figure 1: The structure of our model

The key distinction of our model from AAE is the intervention loss. More specifically, when encoded latent $z = E(x)$ is obtained, for every $k$, $0 \leq k \leq t$, we intervene $z$ by $T_k$ to get $z_k$. Then we feed $z_k$ into the discriminator to obtain a $t + 1$ size output $d$, and compute the cross entropy between $d$ and labels $e_k$, averaging on $k$, finally to update discriminator by gradient flow. The encoder update in the same manner.

Therefore the intervention loss is an unbiased estimator of its theoretical form as follow:

$$\mathcal{L}_{IV}(E, D) = -\mathrm{E}_{k \sim \mathcal{U}[t]}\mathrm{E}_{x \sim p_{\mathrm{data}}}[\log(D_k(T_k(E(x))))]$$
$$= -\mathrm{E}_{k \sim \mathcal{U}[t]}\mathrm{E}_{z \sim p_k}[\log(D_k(z))] \tag{1}$$

where $D_k$, $p_k$ represent the $(k+1)$th digit of $D$ output, the distribution of $z_k = T_k(E(x))$, respectively.

Then we give theoretical results that evidence the rationality of our proposed models.

**Theorem 2** *Given $t + 1$ latent variables $\{z_k\}_{k=0}^t$ with density function support in $R^d$, and their probabilistic density function $\{p_k\}_{k=0}^t$, the classifier $D$ is trained to minimize the cross entropy loss as equation 1 under the constraint that $\sum_{i=0}^t D_j(z) = 1$. Then the optimal classifier satisfies:*

$$D_i^*(z) = \frac{p_i(z)}{\sum_{j=0}^t p_j(z)}$$

*And the corresponding optimal intervention loss can be expressed as follows:*

$$\mathcal{L}_{IV}(E, D) = -JS(p_0, p_2, ..., p_t) + \log(t + 1)$$

*where $JS(q_1, q_2, ..., q_k) = \frac{1}{k}\sum_{j=1}^k (KL(q_j||q))$, $q = \frac{1}{k}\sum_{j=1}^k q_j$.*

*Proof*: See Appendix.

Theorem 2 tells us that when the discriminator reaches optimal, the intervention loss can be regarded as a certain distance among multiple distributions. We know when $t = 1$, the loss becomes the original $JS$ divergence which conforms with GAN. However, it is known that $JS$ divergence isn't a perfect measure because the gradient would be zero if two variables have distributions with disjoint support. According to the theorem, situation for multiple distributions will suffer less from such problem, for bridge distributions provide greater chances for intersection which consequently produces gradient. Therefore it would be more stable to update the encoder.

We now further illustrate the advantage of the multi-object adversarial training. Although GAN's convergence has been theoretically proved(Goodfellow et al., 2014), the training process, is well-known delicate and unstable. One important reason is that the optimal discriminator can always achieve too perfect no matter how close it is between the generated data manifold and the real data manifold (Arjovsky & Bottou, 2017). One cause for this is that two manifolds is disjoint with probability 1 due to their low dimension(Arjovsky & Bottou, 2017) on the data space.

---

**Algorithm 1** Intervention Adversarial Auto-Encoder (IVAAE)

---

**Input:** learning rate: $\alpha$, dimension of the space: $d$, number of bridge distributions: $t$ Interventions: $T_k$, $0 \leq k \leq t$, hyperparameters: $\lambda$, $\mu_1$, $\mu_2$, minibatch size: $m$.
**Output:** $\theta_D, \theta_E, \theta_G$

1: **for** number of training iterations **do**
2:     Sample $\{x_j\}_{j=1}^m$ from the training set
3:     Sample $\{z_j\}_{j=1}^m$ from the prior P(z)
4:     Compute $\hat{z}_j = E(x_j)$, $j = 1, ..., m$
5:     Compute $\hat{x}_j = G(\hat{z}_j)$, $j = 1, ..., m$
6:     **for** $k = 0, ..., t$ **do**
7:         Compute the intervened latent variables
8:         $\hat{z}_{j_k} = T_k(\hat{z}_j)$, $j = 1, ..., m$
9:     **end for**
10:    Update the parameters of $D$ by:
11:    $\theta_D \leftarrow \theta_D + \frac{\alpha}{mt}\mu_1 \nabla_{\theta_D} \sum_{j=1}^m \sum_{k=0}^t \log D_k(\hat{z}_{j_k})$
12:    Calculate $\hat{\mathcal{L}}_{recon}, \hat{\mathcal{L}}_{IV}$
13:    Update the parameters of $G$ by:
14:    $\theta_G \leftarrow \theta_G + \frac{\alpha}{m}\nabla_{\theta_G}\lambda\hat{\mathcal{L}}_{recon}$
15:    Update the parameters of $E$ by:
16:    $\theta_E \leftarrow \theta_E + \frac{\alpha}{m}\nabla_{\theta_G}\{\lambda\hat{\mathcal{L}}_{recon} + \mu_2\hat{\mathcal{L}}_{IV}\}$
17: **end for**
18: **return** $\theta_D, \theta_E, \theta_G$

---

Different from GAN, AAE applies discriminator on the latent distribution, therefore what it learns isn't a low-rank manifold, but a continuous distribution with full dimension. But the problem doesn't get solved in AAE setting. We address it in a heuristic way. We run AAE on real-world dataset for 50 epochs, then fix the encoder and decoder and update the discriminator alone. We observe how perfect the discriminator will develop. Figure 2 records the changing curve of adversarial loss every iteration. According to Figrue 2, the adversarial loss still decreases remarkably toward 0, especially when latent dimension goes high. That's because of the limit batch size. IVAAE could alleviate this problem by boosting the difficulty for discriminator to classify more than two distributions and equivalently introducing more particles. We apply our models IVAAE. We choose $t = 4$, block-wise substitution, with the rest of structure all the same as the AAE model. The loss is recorded with fixed encoder and decoder after 50 epochs, too. We note that although the adversarial loss is decreasing as well, the drop is much more moderate than AAE. We also examine the consequence of such loss decline. We choose the adversarial loss for encoder: $-\mathrm{E}_{p_{\mathrm{data}}}[\log(D(E(x)))]$, and record the gradient norm of the encoder after the encoder and decoder fixed. According to Figure 3, AAE is less stable compared to IVAAE.

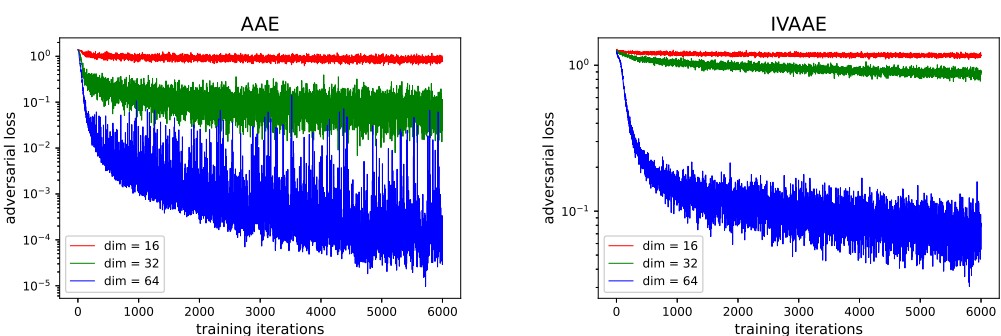

Figure 2: We train the model for 50 epochs. Left: AAE; Right: IVAAE ($t = 4$, block-wise). Then we fix the encoder and decoder to run the discriminator alone. We record the loss change every iteration. The dimension of latent space is chosen at 16, 32, 64.

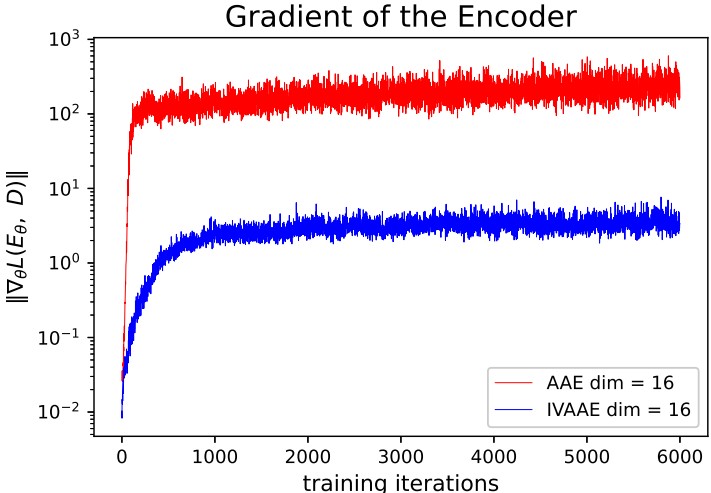

Figure 3: We choose the latent dimension at 16 and run models for 50 epochs. Then we record the change of encoder's gradient with the encoder and decoder fixed. Note that $\mathcal{L}_{adv}(E, D) = -\mathrm{E}[\log(D(E(x)))]$ is used as the encoder loss.

## 3 RELATED WORK

### 3.1 GENERATIVE MODELS

Since the appearance of VAE and GAN model, large amount of attempts have been conducted to improve their performance on generating data. Many of them put their attention on the latent space. 2-stage VAE(Dai & Wipf, 2019) diagnoses the poor generate quality of VAE model and attributes it to the mismatch of the distribution between the encoded $q(z)$ and the prior $p(z)$. They handles the problem by imposing another VAE structure to learn the latent distribution, which improves the model's sample quality. RAE(Ghosh et al., 2019) view VAE from a deterministic angle. It abandons the stochastic component and replace it with new proposed regularization term, which provide an alternative to encode smooth and meaningful latent variables. Info VAE(Zhao et al., 2019) enhance VAE model by introducing to ELBO a mutual information term between $x$ and $z$, which is a generalization of both VAE and AAE. It improves the quality of the variational posterior and strengthen the bond between data space and latent features. Our method, different from the above, dedicates to improve regularity of latent space by enhancing and stabilizing the training process. It applies solid schemes and techniques while not deviating from the original target.

### 3.2 SVGD

Stein variational gradient descent (SVGD) is proposed as a variational inference algorithm (Liu & Wang, 2016). The method iteratively transports a set of particles to match the target distribution by applying a form of functional gradient descent that minimized KL divergence among a reproducing kernel Hilbert space (RKHS). By using the assignment as follow:

$$z_i^{l+1} \leftarrow z_i^l + \frac{\epsilon}{m} \sum_{j=1}^{m} [k(z_j^l, z) \nabla_{z_j^l} \log p(z_j^l) + \nabla_{z_j^l} k(z_j^l, z)], \qquad (2)$$

where the $k$ represents the kernel of RKHS, $m$ batch size, the particles $z_i$ gradually approximate the target distribution $p(z)$. We apply the method in the construction of bridge distribution of the IVAAE model. We choose step size $\epsilon = 1$, kernel function RBF $k(z, z') = \exp(-\frac{1}{\mathrm{h}} \|\mathrm{z} - \mathrm{z}'\|^2)$ with bandwidth $h$ , and set the target distribution to the prior distribution $p(z)$, which is standard

normal in our context. To calculate the intervention loss, we obtain the batched encoded data $z_i = E(x_i), i = 1 \ldots m$. We plug it in the iterative function above to update it $t$ times to get $\{z^l\}_{l=1}^t$. Then we use $\{z^l\}$ as the input of discriminator to obtain the loss term and do the back propagation all the way back. This is computationally practical because of the trivial form of $\nabla \log p(z)$.

## 4 EXPERIMENTS

In this section we conduct experiments on multiple datasets including MNIST (LeCun, 1998), CI-FAR10 (Krizhevsky & Hinton, 2010), CelebA (Liu et al., 2018). We use different patterns of bridge distribution in IVAAE and compare the performance to those of the related baseline method with similar architecture. By exhibiting multiple measures during the train and showing the generated pictures, we empirically demonstrate that the performance of our methods is superior on training stability and generating images quality. The measures we use include FID (Heusel et al., 2017), a popular index to measure the generating variety and quality, and *Frechet Latent Distance* (FLD), a similar form of distance we propose to judge the latent distribution. The bridge distributions we employ include blockwise and radial patterns, corresponding method denoted by $\text{IVAAE}_{bw}$ and $\text{IVAAE}_{rad}$, respectively. We choose $t = 4$ in all of our experiments. The competing approaches we apply include WAE-GAN, for it generalizes the AAE algorithm, and WAE-MMD as well.

We use Pytorch to implement our models. In IVAAE and other baseline models we adopt similar architecture in order to give persuasive conclusion. We use deterministic models only. For all datasets and models, we set hyperparameters as follows: learning rate $\alpha = 0.001$ for encoder and decoder, $\alpha = 0.0005$ for discriminator, $t = 4$, $\lambda = 1$, $\mu_1 = \frac{4}{t+1}$, $\mu_2 = \frac{2}{t+1}$, learning rate decays by multiplying 0.5 after 30 epochs, further multiplying 0.2 after 50 epochs, and 0.1 after 100 epochs. We run all the codes for 300 epochs. Note that the intervention term in IVAAE is down weighted by $\frac{c}{t+1}$ because the derivatives of logrithmatic function becomes $\frac{2}{t+1}$ greater when striking balance at $\frac{1}{t+1}$, instead of $\frac{1}{2}$ in binary conditions. There's no noise added in any layers.

We calculate FID by generating 20k latent variables from the prior and feeding them to the trained decoder. We randomly choose 60k images, get the encoded z, and use the Frechet distance to measure its difference with the standard Gaussian:

$$\text{FLD}(z) = \|\mu_z\|^2 + \text{Tr}(\Sigma_z - 2\sqrt{\Sigma_z} + I)$$

where $\mu_z$ and $\Sigma_z$ is the means and covariance matrix of $z$, respectively. We choose the model with the lowest FID score and calculate its FLD score to measure the regularization performance of latent variables.

**MNIST**

The MNIST dataset contains 60K images of $28 \times 28$ handwritten digits. Before we use them, we resize the images to $64 \times 64$ pixels. Instead of going for the best model performance, we adopt the same architecture as WAE (Tolstikhin et al., 2017) do to compare the results among those models with same structure. We choose the latent size 16, batch size 64. The network structure is two fully connected linear layers for encoder, decoder and discriminator. The only difference in WAE and IVAAE's model structure is the output size of discriminator. See Table 3 in appendix. We use learning rate distinct with the ground settings: 0.0001 for decoder and discriminator, 0.00005 for the encoder. We run each model for 300 epochs and learning rate is divided by 2 every 100 epochs.

**CelebA**

CelebA dataset contains over 200k images of humans faces. We apply transform to crop $140 \times 140$ pixels from the central of images and then resize to $64 \times 64$ digits before we use them. The architecture we adopt is similar to what is recommended in WAE. The dimension of latent space is 64. We use convolutional network for encoder and decoder with $5 \times 5$ or $6 \times 6$ size of filters, and a network composed of four fully connected linear layers of 512 elements for the discriminator. See Table 4 in appendix. Batch size is set at 100.

**CIFAR10**

To implement methods on CIFAR10 we set dimension of latent space 128. We reshape the CIFAR10 images to $64 \times 64$ size. The network structure is completely the same as the network for CelebA, except for the output size of encoder and input size for decoder and discriminator. The latent dimension is relatively large considering batch size we choose as 128. We will note that WAE-GAN fails to converge after 30-40 epochs while IVAAE is stable during the whole training 4.

Except for block-wise replacement and weighted Gaussian, we also use the SVGD(Liu & Wang, 2016) technique as a third method to construct bridge distribution. We obtain particle assembles $z^l$ from encoded batched data $z$ by iterative assignment (equation 2). A notable question is, once the batch size is limited, the particle assemble $z^l$ is not approaching standard normal but with a shrunk variance lower than $1$. Therefore we need a high width h which could enlarge the repulsive force between points. The discrepancy is also amplified when the data's dimension goes higher and batch size goes smaller. Therefore we implement the method only on MNIST dataset which needs relatively low latent dimension.

We let $t = 4$ and choose RBF kernel. The step size $\epsilon$ is set to 1. We obtain $\{z^l\}_{l=1}^{t-1}$ for every two iterations, while $z^t$ is sampled from standard Gaussian. SVGD degrades to a gradient descent of MAP when width $h = 0$, while high $h$ would cause the convergence to slowing down, so finally we choose $h = 100h_0$, $h_0 = \frac{\text{med}^2}{\log n}$ as recommended (Liu & Wang, 2016), where $\text{med}$ is the median of the pairwise distance between the current points $z_l$. Besides, we add 0.1 Gaussian noise on the particles after every iteration.

Note that in SVGD process points update in a relative smooth way by adding deviation on former points. It is similar to the radial substitution. Moreover, when using equation 2 to update a latent point, all points in current batch are involved, therefore the tendency of one point towards Gaussian distribution is directly affected by force from other points. Such interaction become stronger under a great bandwidth $h$. Intuitively we hope this feature would improve the hidden layer's regularity. To measure such regularity to a certain extent, we utilize the trained encoder to do classification work. We fix the parameters of encoder after training WAE-GAN, IVAAE$_{rad}$, IVAAE$_{svgd}$ models, respectively. Then a two-layer MLP with 1000 nodes each is connected to the encoder's last hidden layer to do supervised learning. We set batch size 60, learning rate 0.0001, and evaluate the classification error rate every 100 training iterations. The result is favorable to our argument as shown in Figure 5:

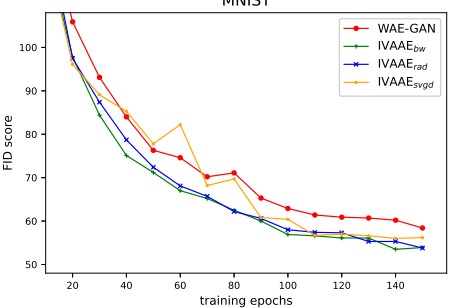 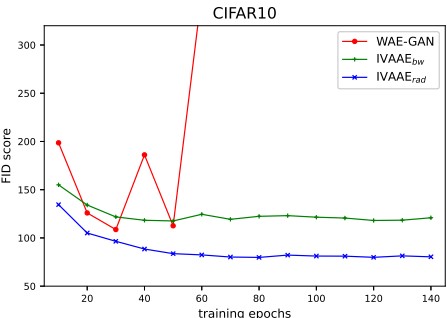

Figure 4: FID curves for different methods calculated during the training. Note that WAE-GAN's training process is not stable on CIFAR10.

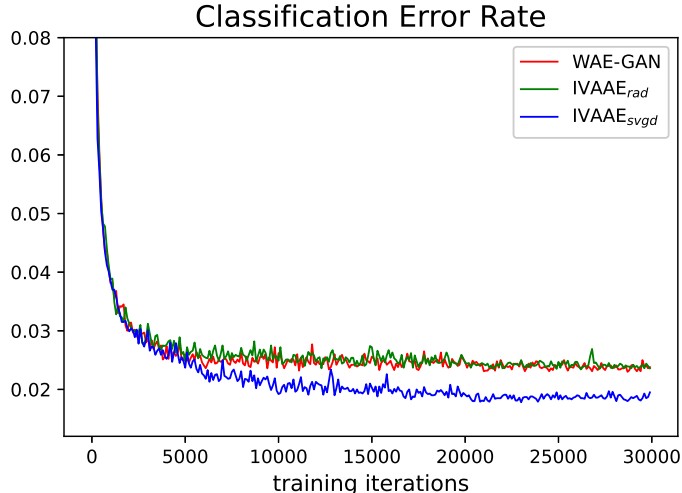

Figure 5: MNIST classification error rate for methods WAE-GAN, IVAAE$_{rad}$, IVAAE$_{svgd}$. Parameters of the encoder is fixed after pretrained.

Table 1: FID score for different methods on multiple datasets. FID results are calculated every 10 epochs and we choose the minimum. Lower score means generated images are better. * means we don't go for the best performance across different structure, but for comparison purpose only.

| Methods | MNIST | CelebA | CIFAR10 |
|---|---|---|---|
| WAE-MMD | 67* | 36 | 87 |
| WAE-GAN | 53* | 33 | 109 |
| IVAAE$_{bw}$ | 49* | 34 | 117 |
| IVAAE$_{rad}$ | 50* | 33 | 80 |
| IVAAE$_{svgd}$ | 49* | - | - |

Table 2: FLD score for different methods on multiple datasets. We choose FLD when FID reaches the best. Lower score means the encoded latent distribution is closer to the Guassian prior.

| Methods | MNIST | CelebA | CIFAR10 |
|---|---|---|---|
| WAE-MMD | 0.05 | 0.34 | 10.9 |
| WAE-GAN | 5.5 | 1.5 | 15.1 |
| IVAAE$_{bw}$ | 0.2 | 0.22 | 4.0 |
| IVAAE$_{rad}$ | 0.07 | 0.18 | 11.8 |
| IVAAE$_{svgd}$ | 5.7 | - | - |

From the result we could notice that WAE-GAN's training stability and speed is lower than IVAAE on dataset we study. It conforms to what have been discussed in previous section. When the latent size reach higher as 128 (with batch size 128), WAE-GAN suffers from intense oscillation. Every time the model converges to a certain degree, the adversarial loss decreases and reconstruction loss increases sharply. IVAAE performs well under the situation we inspect, and the FID score and the FLD [Figure 6] when FID reaches the lowest is overall better [Table 1, 2]. Curiously, in terms of the performance, among patterns we choose there seems to be no pattern that are superior to the other on every datasets. On CIFAR10 we notice that IVAAE$_{rad}$ attain far lower FID score than IVAAE$_{bw}$, while the FLD score goes to the opposite. There seems to be a trade-off relation between the generate quality and the regularity (how close to the standard normal) of latent variables. It is an interesting phenomenon to explore in the future work.

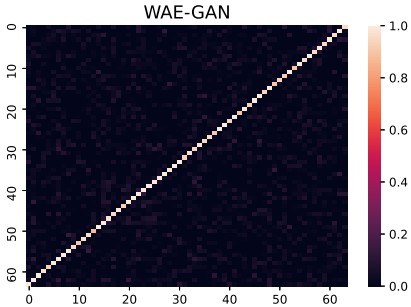 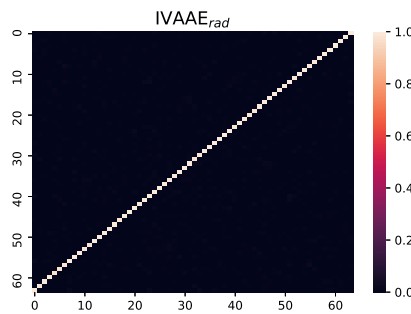

Figure 6: Heat maps for covariance matrix of the encoded $z$ for different models when converging. We choose CelebA with the latent dimension as 64. The brightness represent absolute value of the corresponding entry. The brighter the closer to 1. The darker the closer to 0.

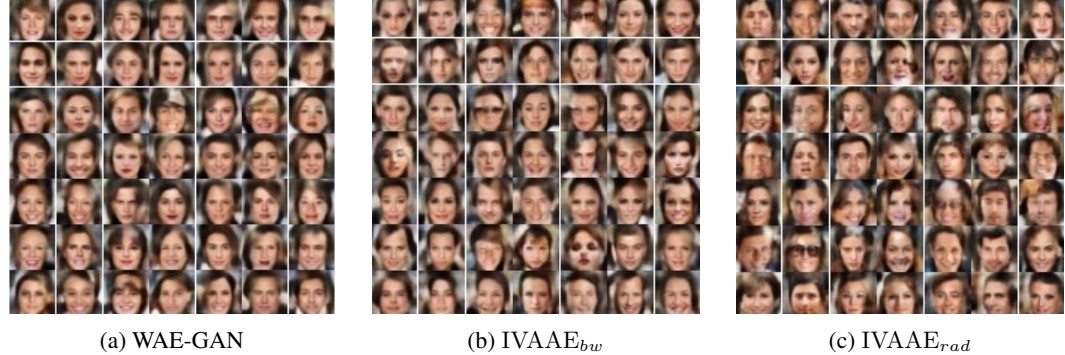

(a) WAE-GAN          (b) IVAAE$_{bw}$          (c) IVAAE$_{rad}$

Figure 7: CelebA images randomly generated. Left: WAE-GAN; Mid: IVAAE$_{bw}$; Right: IVAAE$_{rad}$.

## 5   CONCLUSION

In this paper we propose a generative model which possess robust and stable training property by leveraging interventions on latent variables and classifying them. It creatively applies the multiple-output structure to discriminate over two distributions. It is theoretically proved that the intervened latent distributions can be regarded as certain bridge distributions which work as a guidance to pull the encoded variables to the target distribution. By conducting a heuristic loss trace for discriminator, we demonstrate that multiple classifying have a significant stabilizing effect on the adversarial training in practice.

To structure different bridge distribution, we also apply the SVGD method to smoothly update the latent variables. We successfully combine the method in our adversarial training, and to some extent demonstrate the hidden space enjoys better regularity as well.

Besides, multiple experiments are conducted on real-world datasets. By using measures on both images generating quality and latent regularity, we conclude a better performance of our models compared to the baseline models. The data results coincide with our inference on the stability of model training.

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

## A Appendix

### A.1 Proof of Theorem 2

$$\mathcal{L}_{IV}(E, D) = -\mathbb{E}_{k \sim \mathcal{U}[t]}\mathbb{E}_{z \sim p_k}[\log(D_k(z))]$$

$$= -\frac{1}{t+1}\sum_{k=0}^{t}\int_z p_k(z)\log D_k(z)dz$$

$$= \int_z -p(z)\sum_{k=0}^{t}p(e_k|z)\log D_k(z)dz$$

Last equation holds because $p(e_k|z) = \frac{p(e_k)p_k(z)}{p(z)} = \frac{p_k(z)}{(t+1)p(z)}$.
Notice that

$$-\sum_{k=0}^{t}p(e_k|z)\log D_k(z) = -\sum_{k=0}^{t}p(e_k|z)\log\frac{D_k(z)}{p(e_k|z)} + \mathrm{H}(p(\cdot|z))$$

$$\geq -\log\sum_{k=0}^{t}p(e_k|z)\cdot\frac{D_k(z)}{p(e_k|z)} + \mathrm{H}(p(\cdot|z))$$

$$= \mathrm{H}(p(\cdot|z))$$

The equality holds when $D_k^*(z) \propto p(e_k|z) \propto p_k(z)$, i.e. $D_k^*(z) = \frac{p_k(z)}{\sum_{j=0}^{t}p_j(z)}$.
Plugging $D^*(z)$ into the loss term, we obtain:

$$\mathcal{L}_{IV}(E, D) = -\frac{1}{t+1}\sum_{k=0}^{t}\int_z p_k(z)\log D_k(z)dz$$

$$= -\frac{1}{t+1}\sum_{k=0}^{t}\int_z p_k(z)\log\frac{p_k(z)}{\sum_{j=0}^{t}p_j(z)}dz$$

$$= -\frac{1}{t+1}\sum_{k=0}^{t}\int_z p_k(z)\left[\log\frac{p_k(z)}{p(z)} - \log(t+1)\right]dz$$

$$= -\frac{1}{t+1}\sum_{k=0}^{t}KL(p_k\|p) + \log(t+1)$$

So we have our theorem proved. $\square$

### A.2 Network Structure

Table 3: The network architecture we use for MNIST. FC represents fully connected layer.

| E | G | D |
|---|---|---|
| INPUT $64 \times 64$ | INPUT $z$ | INPUT $z$ |
| FC(1024) | FC(16, 1024) | FC(16, 1024) |
| ReLU | ReLU | ReLU |
| FC(1024, 1024) | FC(1024, 1024) | FC(1024, 1024) |
| ReLU | ReLU | ReLU |
| FC(1024, 16) | FC($64 \times 64$), Tanh | FC($t+1$) |

Table 4: The network architecture we use for CelebA and CIFAR10. CONV(C, K, S) represents convolutional layer with C channels, K×K-size kernels, and S strides. DCONV represents deconvolutional layer, parameter meaning similar with CONV. BN represents batch normalization layer.

| E | G | D |
|---|---|---|
| INPUT $64 \times 64 \times 3$ | INPUT $z$ | INPUT $z$ |
| CONV(C128, K5, S2) | FC($1024 \times 8 \times 8$) | FC(512) |
| BN, ReLU | DCONV(C512, K5, S1) | ReLU |
| CONV(C256, K5, S2) | BN, ReLU | FC(512, 512) |
| BN, ReLU | DCONV(C256, K5, S1) | ReLU |
| CONV(C512, K5, S2) | BN, ReLU | FC(512, 512) |
| BN, ReLU | DCONV(C128, K5, S2) | ReLU |
| CONV(C1024, K5, S2) | BN, ReLU | FC(512, 512) |
| BN, ReLU | DCONV(C1, K4, S0) | ReLU |
| FC(C1) | Tanh | FC($t + 1$) |

