# OpenReview forum: "Intervention Adversarial Auto-Encoder"
_ICLR.cc/2022/Conference — ICLR 2022 Submitted_

### Official Review · Reviewer_Nq3D · 2021-10-17

**Correctness:** 2
**Technical Novelty And Significance:** 2
**Empirical Novelty And Significance:** 2
**Recommendation:** 3
**Confidence:** 4

**Main Review:**

Strengths:
1. The paper targets on several serious concerns about generative modeling, such as unstable training process and mode collapse.

Weaknesses:
1. The novelty of this paper is not significant enough as similar ideas have been developed for GAN in "Jiadong Liang, Liangyu Zhang, Cheng Zhang, and Zhihua Zhang. Intervention generative adversarial networks."
2. The experiment evaluation is not sufficient as the only comparing method is WAE. There are many methods targeting on unstable training process and mode collapse in GAN, and more comparisons are needed, which should also be included in related work and reference.
3. The improvement shown in experiments is marginal and not significant.
4. As the key component of IVAAE, page 2 shows two substitutions, more systematic discussion is needed on how to develop and assess substitutions.
5. On page 4, "We address it in a heuristic way." Detailed settings for this heuristic study should be provided, why can this particular setting be representative and generalize to other settings?

**Summary Of The Paper:**

This paper presents Intervention Adversarial auto-encoder to stabilize the training process of the latent variables of adversarial auto-encoders. Intervention Adversarial auto-encoder adopts a sequence of distributions to bridge the distribution of the learned latent variable and its prior distribution. The bridge distributions are implemented by a multi-output discriminator, which guides the initial latent distribution to the target distribution with a stable training process. The paper discusses several different types of the bridge distributions and applies
Stein variational gradient descent. Experiments on multiple real-world datasets are conducted.

**Summary Of The Review:**

This paper targets on some important issues in generative modeling such as unstable training process and mode collapse. Novelty of this paper is low and evaluation is not sufficient.

---

### Official Review · Reviewer_tnv3 · 2021-10-30

**Correctness:** 2
**Technical Novelty And Significance:** 2
**Empirical Novelty And Significance:** 2
**Recommendation:** 3
**Confidence:** 5

**Details Of Ethics Concerns:**

I have no ethical concerns about this paper.

**Main Review:**

## Strength:
The intervention loss for regularization seems novel in VAE training. But it's far below the bar of ICLR. I will describe the weakness as:

## Weakness:
1. Lots of related works are missing in this paper, e.g. ARAE [1] and VAE-SVGD [2].

2. Though the method is motivated well, the proposed method doesn't seem to solve the problem of VAE, especially considering only WAE as baselines.

3. The experiments are weak. Recall the baseline works that have been published several years ago, this paper is not even close to them.

[1] Adversarially Regularized Autoencoders

[2] VAE Learning via Stein Variational Gradient Descent

**Summary Of The Paper:**

This paper introduces an intervention adversarial autoencoder and claims this could improve VAE learning.

**Summary Of The Review:**

I would suggest the authors choose another conference.

---

### Official Review · Reviewer_LAu5 · 2021-11-02

**Correctness:** 2
**Technical Novelty And Significance:** 2
**Empirical Novelty And Significance:** 2
**Recommendation:** 3
**Confidence:** 3

**Main Review:**

Pros
+ The topic is interesting.

Cons
- The technical correctness is very concerning. There are many claims left unsupported (via references or experiments) or poorly explained at best. The most serious ones are about stability claims, including "according to Figure 3, AAE is less stable compared to IVAAE" on page 4 and "we propose a generative model which possess robust and stable training" on page 9. The only result which very weakly (given the limited scale of experiments) suggests better stability in the paper is the CIFAR10 panel in Figure 4, which is clearly insufficient to support the aforementioned claims. Also, Algorithm 1 has many forward and backward (backprop) passes (some likely redundant), but the speed (based on wall-clock time) is never discussed in the paper.
- The novelty is quite poor. The intervention loss (Liang et al., 2020) is only slightly improved by introducing the radial substitution, which is very close to mixup in the latent space [1,2]. The addition of SVGD is somewhat random and poorly justified. Given that Gaussianization has been studied for a long time [3,4,5], it's unclear why the much more generic SVGD should be used instead. The "proposed" FLD is simply just [6] and not helpful in explaining the results (since it's uncommon). Normality tests [7] which can directly give probabilities would be much better in my opinion.
- The significance is quite low. The scale of the experiments and the number of baselines are simply too small to convince the research community that the proposed algorithms are indeed better and the results will generalize. The FID scores and the quality of the generated images are also clearly too low for today's standards. Theorem 2 is basically reiterating (Liang et al., 2020).
- The writing is poor and hard to follow given the amount of errors. Also, Eq 2 is clearly missing some subscripts. Fig 4 and 7 are not referenced in the maintext. AAE was authored by Alireza Makhzani, Jonathon Shlens, Navdeep Jaitly and Ian Goodfellow, not just Makhzani and Shlens.

[1] On Adversarial Mixup Resynthesis, NeurIPS, 2019.\
[2] Manifold mixup: Better representations by interpolating hidden states, ICML, 2019.\
[3] Gaussianization, NeurIPS, 2000.\
[4] Iterative gaussianization: from ica to random rotations, IEEE TNN, 2011.\
[5] Gaussianization flows, AISTATS, 2020.\
[6] The Fréchet distance between multivariate normal distributions, Journal of Multivariate Analysis, 1982.\
[7] https://en.wikipedia.org/wiki/Normality_test

**Summary Of The Paper:**

The paper aims to improve AAEs with the intervention loss (Liang et al., 2020) and SVGD (by constructing "bridge distributions") and demonstrates some empirical results.

**Summary Of The Review:**

Given the concerning technical correctness, poor novelty, significance and writing, I recommend to reject this paper.

---

### Decision · Program_Chairs · 2022-01-20

**Decision:**

Reject

**Comment:**

This paper aims at improving AAEs with an intervention loss. While the topic is important, the reviewers agree that

- The paper has poor clarity,
- The related work is not adequately put into perspective,
- There are concerns with technical correctness,
- Experimental evidence is lacking,

As the authors have not addressed any of these concerns, the paper can not be accepted in its current form.